# The First Five Mitochondrial Genomes for the Family *Nidulariaceae* Reveal Novel Gene Rearrangements, Intron Dynamics, and Phylogeny of *Agaricales*

**DOI:** 10.3390/ijms241612599

**Published:** 2023-08-09

**Authors:** Zhao-chen Li, Tian-chen Xie, Xi-long Feng, Zhen-xin Wang, Chao Lin, Guo-ming Li, Xiu-Zhang Li, Jianzhao Qi

**Affiliations:** 1Shaanxi Key Laboratory of Natural Products & Chemical Biology, College of Chemistry & Pharmacy, Northwest A&F University, Yangling, Xianyang 712100, China; 2State Key Laboratory of Plateau Ecology and Agriculture, Qinghai Academy of Animal and Veterinary Sciences, Qinghai University, Xining 810016, China

**Keywords:** *Nidulariaceae*, mitochondrial genomes, phylogenetic analysis, *Agaricales*

## Abstract

The family *Nidulariaceae*, consisting of five genera including *Cyathus*, is a unique group of mushrooms commonly referred to as bird’s nest fungi due to their striking resemblance to bird’s nests. These mushrooms are considered medicinal mushrooms in Chinese medicine and have received attention in recent years for their anti-neurodegenerative properties. However, despite the interest in these mushrooms, very little is known about their mitochondrial genomes (mitogenomes). This study is the first comprehensive investigation of the mitogenomes of five *Nidulariaceae* species with circular genome structures ranging in size from 114,236 bp to 129,263 bp. Comparative analyses based on gene content, gene length, tRNA, and codon usage indicate convergence within the family *Nidulariaceae* and heterogeneity within the order *Agaricales*. Phylogenetic analysis based on a combined mitochondrial conserved protein dataset provides a well-supported phylogenetic tree for the Basidiomycetes, which clearly demonstrates the evolutionary relationships between *Nidulariaceae* and other members of *Agaricales*. Furthermore, phylogenetic inferences based on four different gene sets reveal the stability and proximity of evolutionary relationships within *Agaricales*. These results reveal the uniqueness of the family *Nidulariaceae* and its similarity to other members of *Agaricales*; provide valuable insights into the origin, evolution, and genetics of *Nidulariaceae* species; and enrich the fungal mitogenome resource. This study will help to expand the knowledge and understanding of the mitogenomes in mushrooms.

## 1. Introduction

Mitochondria are essential organelles in fungi, and they play crucial roles in the growth, development, and adaptation of fungal species [1,2]. The fungal mitochondrial genome (mitogenome) is typically made up of a circular DNA that encodes fourteen mitochondria special protein-coding genes (PCGs), ribosomal RNAs, and transfer RNAs, which are usually found in clusters [3,4]. The size of fungal genomes varies greatly [4], ranging from 19,431 bp in *Schizosaccharomyces pombe* [5] to 332,165 bp in *Golovinomyces cichoracearum* [6]. Mitochondrial genome size varies considerably among and within fungal species [7,8], and the principal drivers of this variation are the presence and proliferation of introns within the protein-coding genes, as well as significant non-coding regions (NCRs) [3,9]. To gain insight into the evolution of fungal mitochondrial genomes, it is -critical to investigate the dynamics of introns and determine the origin and function of these NCRs [10]. The mitochondrial genomes (mitogenomes) of fungi are valuable tools for studying the evolutionary history and phylogenetic relationships due to their small sizes, conserved orthologous genes, high copy numbers, low recombination rates, and high evolutionary rates [11,12,13,14]. In fact, the mitogenome is often referred to as the “second genome” of eukaryotes due to its significant contribution to understanding eukaryotic individual growth and environmental adaptation [15,16]. Additionally, studying the mitogenomes of pathogenic fungi can provide insight into the mechanisms underlying their pathogenicity in plants [17,18]. Despite its importance, the fungal mitochondrial genome has received relatively little attention. To date, fewer than 1300 fungal mitochondrial genomes have been deposited in the NCBI database (https://www.ncbi.nlm.nih.gov/genome/browse#!/organelles/, accessed on 1 January 2023), and this number is far less than the number of published fungal nuclear genomes and particularly the number of DNA parts.

Mushrooms, the common name for macrofungi capable of forming specific fruiting bodies, have emerged as a significant source of food and medicine for humans due to their nutritional value and rich content of bioactive compounds [19,20,21]. As the importance of medicinal mushrooms for human health continues to grow and sequencing technologies become increasingly available, the genomes of valuable medicinal mushrooms are being sequenced and deciphered [22,23,24]. However, the mitogenomes of these mushrooms remain relatively understudied. Among the well-known medicinal mushrooms in Chinese medicine, *Cyathus* species have garnered much attention in recent years for the production of cyathane diterpenoids with significant neurotrophic and neuroprotective activities [25,26,27,28,29]. The family *Nidulariaceae*, commonly known as “bird’s nest fungi”, is a distinctive and fascinating group of fungi in the fungal kingdom [30], comprising five genera: *Cyathus*, *Crucibulum*, *Mycocalia*, *Nidularia*, and *Nidula* (Figure 1). Although the family *Nidulariaceae*, represented by the genus *Cyathus*, is classified in the order *Agaricales*, there are obvious morphological differences between the family *Nidulariaceae* and other mushrooms of the order *Agaricales* (Figure 1). To date, 30 mitogenomes have been reported from the order *Agaricales* [31], but none have been reported from the family *Nidulariaceae*. In addition to the obvious differences in morphology, whether there are also obvious differences in the mitogenomes of *Nidulariaceae* and other *Agaricales* mushrooms is an intriguing scientific question that deserves attention and investigation.

The present study aimed to investigate the mitogenomes of five *Cyathus* species, namely *C. striatus* 87405, *C. striatus* AH44044, *C. jiayuguanensis* 765, *C. pallidus* QL1, and *C. stercoreus* NPCB004. This is the first report of *Nidulariaceae* mitogenomes and allowed the characterization of these genomes, including their base composition, gene distribution, intron dynamics, codon usage bias, and PCG characteristics. Comparative analyses were performed within the order *Agaricales* to explore the correlation between genome size and the proportion of unannotated open reading frames, homing endonucleases, intronic regions, and intergenic regions. The results showed a positive correlation between genome size and these factors. Genetic distance and evolutionary selection pressure (Ka/Ks) analyses indicated that the *nad6* and *rps3* were not conserved in the order *Agaricales* and that *rps3* showed species specificity. Furthermore, gene rearrangement analysis revealed uniformity within the genus *Cyathus* and heterogeneity between *Cyathus* and other members of the order *Agaricales*. Phylogenetic inference based on combined sequences of fourteen conserved proteins and four datasets (PCG, PCG12, PCGR, and PCG12R) within *Agaricales* further confirmed the evolutionary position of *Nidulariaceae* within the order and the topology of the *Agaricales* evolutionary tree. This study provides valuable insights into the origin, evolution, and genetics of *Nidulariaceae* species and closely related members within *Agaricales*. The reported mitogenomes provide a valuable resource for future studies of *Cyathus* species and for comparative mitogenomics among Basidiomycetes mushrooms.

## 2. Results

### 2.1. Characterization of the Five Cyathus Mitogenomes

The five *Cyathus* species possessed circular mitogenomes of varying sizes: 115,936 bp for *C. jiayuguanensis* 765, 127,920 bp for *C. striatus* 87405, 120,589 bp for *C. striatus* AH440044, 114,236 bp for *C. pallidus* QL1, and 129,263 bp for *C. stercoreus* NPCB004 (Figure 2). With an average GC content of 28.39%, interspecific variation is evident—the highest being 30.12% in *C. jiayuguanensis* 765 and the lowest being 27.34% in *C. striatus* AH44044. While *C. jiayuguanensis* 765 exhibited negative AT skew (−0.011) and positive GC skew (0.010), the remaining four mitogenomes showed positive AT skews (0.066–0.094) and negative GC skews (−0.147 to −0.122). In total, the mitogenomes contained 43, 44, 39, 31, and 38 protein-coding genes for *C. striatus* 87405 and AH44044, *C. jiayuguanensis* 765, *C. pallidus* QL1, and *C. stercoreus* NPCB004, respectively (Appendix A). Each mitogenome encoded 15 core PCGs, including five cytochrome C oxidase subunit genes, five ATP synthase subunit genes, six NADH dehydrogenase subunit genes, and one ribosomal protein S3 gene. These core PCGs, except for *rps3*, are conserved mitochondrial genes involved in the oxidative phosphorylation pathway [32]. Furthermore, two ribosomal RNAs (rns and rnl) were present in all five *Cyathus* mitogenomes, with the number of tRNA genes ranging from 24 to 28 (Figure 2, Appendix A).

The majority of fungal mitogenomes contained a collection of uncharacterized open reading frames (un_ORFs), which encode either unidentified proteins or misidentified known proteins [33]. Notably, five un_ORFs encoding unknown proteins were identified in *C. striatus* 87405 and AH44044, accompanied by several copies of genes encoding GIY and LAGLIDADG endonucleases. Similarly, *C. jiayuguanensis* 765 harbored un_ORFs and putative reverse transcriptase encoding genes, as well as an abundance of GIY and LAGLIDADG endonuclease gene copies. The remaining two species, *C. pallidus* QL1 and *C. stercoreus* NPCB004, exhibited a limited number of un_ORFs, alongside multiple copies of GIY/LAGLIDADG-related genes (Appendix A). We also observed introns present in the mitochondrial genomes of different species within this genus, with significant differences in their distribution. Intronic analyses revealed that *C. striatus* 87405 and AH44044 contained more introns than other species, distributed in *cob*, *cox1*, *cox3*, *nad1*, *nad5*, and *rnl* genes, whereas *C. jiayuguanensis* 765 had fewer introns, distributed in *atp9*, *cob*, *cox1-3*, *nad1*, *nad5*, and *rnl* genes. Except for *C. pallidus* QL1, all species exhibited group II introns (Appendix A).

The proportions of intergenic regions, PCGs, intronic regions, and RNA regions were comparable across all five mitogenomes. Intergenic regions were the most abundant, averaging 51.94% of the mitogenomes, while RNA regions contributed the least, averaging 6.50%. PCGs comprised an average of 30.37% of the five *Cyathus* mitogenomes, and intronic regions represented 11.20%. There was significant variation in the proportions of PCGs and intergenic regions between species. PCGs comprised an average of 34.69% of the mitogenomes in *C. striatus* 87405, *C. striatus* AH44044, and *C. jiayuguanensis* 765, while they made up 23.88% in *C. pallidus* QL1 and *C. stercoreus* NPCB004. In comparison, the intergenic regions accounted for 46.95% of the mitogenomes in *C. striatus* 87405, *C. striatus* AH44044, and *C. jiayuguanensis* 765, which was 12.46% lower than the proportion in *C. pallidus* QL1 and *C. stercoreus* NPCB004 mitogenomes (Figure 2, Appendix A).

### 2.2. Codon Usage Analysis

To analyze the codon usage patterns in different *Agaricales* mitogenomes, we compared 34 mitogenomes, including 29 published in the database and 5 *Cyathus* mitogenomes assembled by ourselves. The *cox1* genes in 19 *Agaricales* mitogenomes utilized ATG as start codons, while others employed GTG as start codons, with Amanita muscaria, *C. jiayuguanensis* 765, and *Stropharia rugosoannulata* being exceptions, using TTG. Most of the remaining fourteen core PCGs predominantly used ATG as start codons, although the atp6 gene of Marasmius tenuissimus and the *rps3* gene of *S. rugosoannulata* used GTG as start codons. In addition, the *cox2* gene of *C. striatus*, the nad1 gene of Laccaria bicolor, and the *nad4* and *nad6* genes of *S. rugosoannulata* used TTG as start codons. Among the core PCGs of the 34 mitotic genomes examined, TAA was the most prevalent stop codon, followed by TAG (Appendix A).

Subsequent analysis revealed considerable differences in the use of start and stop codons among *Agaricales* species, even among closely related species. For instance, the cox1 gene of *C. jiayuguanensis* 765 used TTG as the start codon, whereas *C. striatus* 87405 and AH44044 employed ATG as the start codon. The *cox3* gene of *C. jiayuguanensis* 765, *C. pallidus* QL1, and *C. stercoreus* NPCB004 adopted TAG as stop codons, while that of *C. striatus* 87405 and *C. striatus* AH44044 used TAA as stop codons. Within Amanita species, the atp6 gene of *A. muscaria* used TAG as the stop codon, while *A. phalloides* and *A. thiersii* utilized TAA as stop codons (Appendix A).

Codon usage has been shown to directly affect the rate and energy required for translation. Genes in mitochondria are believed to favor specific codons to save time and conserve energy for cell growth [34]. Our in-depth analysis revealed that the most commonly used codons of five *Cyathus* species mitogenomes were UUA (for leucine; Leu), followed by AAA (for arginine; Arg). The codon usage bias for different amino acids is essentially the same among the five *Cyathus* species; for example, UGU was the most frequently used codon for cysteine, and GAA was the preferred codon for glutamic acid (Figure 3A, Appendix A). Moreover, the codon usage bias of the *Cyathus* mitogenomes is mainly consistent with that of *Agaricales* species mitogenomes (Figure 3B, Appendix A), which suggests that mitogenomes within the same order tend to share similar codon usage biases. Some exceptions may be attributed to the diversity of un_ORFs in mitotic genomes.

### 2.3. Molecular-Signature-Based Assessment of the Coding Potential of PCGs

AT-rich mitochondrial genomes have been demonstrated to preferentially use AT-rich codons to encode their genes [35]. To evaluate this observation in 34 *Agaricales* species, we analyzed the correlation between AT content, GC content, and the usage frequency of AT-rich and GC-rich codons. Our results revealed a significant positive correlation between GC/AT content and the usage frequency of GC-rich/AT-rich codons (correlation coefficient = 0.85, *p*-value = 1.60 × 10^−10^) (Figure 4A, Appendix A).

We then calculated the GC content of the CDS and intronic regions, and the results demonstrated that the GC content of the intronic regions was significantly different from that of the CDS by the Wilcoxon test (*p*-value = 0.022). In mammals, GC content can be used as a splicing marker because of exons’ low GC content and intronic sequences’ high GC content [36]. Since the GC content of intronic regions (28.35 ± 4.06%) is higher than that of CDS (26.46 ± 2.31%), our results suggest that the GC content of intronic regions may act as a potential splicing signal in *Agaricales* mitogenomes (Figure 4B, Appendix A).

The t-SNE results visually showed that the relative synonymous codon usage (RUSU) value of core PCGs differed from that of un_ORFs. Approximately 19.25% of the core protein-coding genes and 17.45% of the unidentified genes were present on the reverse strand (Appendix A), which might contribute to the differential RSCU values between the two groups tested. Interestingly, the un_ORFs formed only one cluster, whereas the core PCGs formed two clusters, one of which partially overlapped with the un_ORFs cluster (Figure 4C). This suggested the presence of variation in codon usage bias within the same group. A correlation between gene composition in mitogenomes and location on the strand has been reported in several mammals [37], hinting at a similar pattern in *Agaricales* species. Moreover, distinct replication processes on the forward and reverse strands can result in different mutation rates, affecting codon usage [38]. Our t-SNE findings also imply that un_ORFs and conserved genes might experience different selective pressures. Considering the limited number of tRNA gene copies for each amino acid in mitogenomes, codon optimization is expected to enhance mRNA stability and translation rate. Hence, we recommend that molecular biology experiments be carried out to explore the structure and function of un_ORFs further.

### 2.4. Repetitive Sequence Analysis

The mitogenome of organisms is known to contain various types of repetitive sequences, including tandem and interspersed repeats [39,40]. We analyzed the mitogenomes of five *Cyathus* species to identify and characterize their repetitive sequences. A total of 30 repetitive sequences were present in these five *Cyathus* mitogenomes, with the number of repeats ranging from 11 to 13. The size of these sequences varied from 39 to 624 bp, with the largest repeat (624 bp) being identified in *C. jiayuguanensis* 765. This repeat sequence was located in the intergenic region between the genes *cox3* and *orf275*, as well as in the partial region of the rns gene. Another large repeat sequence of 502 bp was identified in the partial region of the fourth exon and third intron of the *nad3* gene of *C. jiayuguanensis* 765. Pairwise nucleotide identities between the five *Cyathus* mitogenomes ranged from 80.54% to 100%, indicating a high degree of conservation among these species. The repetitive sequences accounted for 0.78% to 3.10% of the five *Cyathus* mitogenomes. The highest proportion of repeat sequences was found in *C. jiayuguanensis* 765, while *C. stercoreus* NPCB004 had the lowest content of repetitive sequences (Appendix A).

A total of 1228 tandem repeats were found in the five *Cyathus* mitogenomes, with the number of repeats ranging from 177 to 313. The longest tandem repeat sequence of 952 bp was identified in *C. jiayuguanensis*. Most of the tandem repeats were duplicated once or twice in the five *Cyathus* mitogenomes, with the highest number of duplications (22) observed in the *C. pallidus* QL1 mitogenome. The proportions of tandem repeat sequences gradually decreased in the mitogenomes of *C. jiayuguanensis*, *C. stercoreus* NPCB004, *C. pallidus* QL1, *C. striatus* 87405, and AH44044 and were 19.95%, 15.69%, 15.37%, 11.46%, and 9.76%, respectively (Appendix A). Our results suggest that repetitive sequences, especially tandem repeats, are common features of *Cyathus* mitogenomes.

### 2.5. Genetic Distance and Evolutionary Rates of PCGs

Within the 15 detected core PCGs, the *rps3* gene exhibited the highest median K2P genetic distance in the 34 *Agaricales* mushrooms, preceded by the *nad6* and *nad3* genes. This was an indication that the three genes may have diverged significantly in the course of evolution. Conversely, the *nad4L* gene had the lowest median K2P distance (Figure 5, Appendix A), suggesting high conservation, probably due to its role as a minimal multi-pass membrane protein required for catalysis in the mitochondrial membrane respiratory chain NADH dehydrogenase (Complex I).

For the 15 scrutinized core PCGs, the *rps3* gene had the highest observed Ka value, while the *nad4L* gene demonstrated the lowest Ka value, parallel to the K2P patterns. The synonymous substitution rate (Ks) of the *cox1* gene was the highest, while that of the *rps3* gene was the lowest among the 34 *Agaricales* species. The Ka/Ks values of 15 core PCGs were <1, indicating that these genes were under purifying selection pressure. In general, a higher *rps3* ratio is more likely with greater species diversity in the Ka/Ks calculation. Intriguingly, the average Ka/Ks value for *rps3* from the 34 *Agaricales* species was 0.82 (Figure 5, Appendix A), which showed that the *rps3* gene may not have been under positive selection and that it is still in the process of phylogenetic differentiation. In conclusion, our findings illuminate the evolutionary dynamics of core PCGs in *Agaricales* species and emphasize the pivotal role of purifying selection in maintaining structural integrity and biological function.

### 2.6. Intron Dynamics of cox1 Genes

A total of 535 introns were found in 34 mitogenomes of *Agaricales* mushroom, located in 12 genes: *atp9*, *cob*, *cox1*, *cox2*, *cox3*, *nad1*, *nad2*, *nad4*, *nad5*, *nad6*, *rns*, and *rnl*. Among them, the *cox1* gene harbored the most mitochondrial introns, with 236 introns, representing 44.11% of all introns in 34 mitogenomes (Figure 6, Appendix A). As a result, the variation in introns in the *cox1* gene could significantly affect the length and structure of mitochondria. The *cox1* gene of *C. striatus* 87405 and AH44044, *C. jiayuguanensis*, *C. pallidus* QL1, and *C. stercoreus* NPCB004 contained 15, 12, 6, 8, and 11 Pcls, respectively, indicating that *Cyathus* species have experienced a conspicuous loss/gain of introns during evolution.

Position class (Pcl) is a term used to define the exact location of the coding region in the *cox1* gene [41], typically used to characterize the positional information of introns contained in the *cox1* gene. Two hundred thirty-six introns detected in the *cox1* genes of 34 *Agaricales* mushrooms were categorized into 43 Pcls based on the sequence comparison of the *cox1* gene with a reference species, *Ganoderma calidophilum* [42]. Discrepancies in the class and quantity of introns in 34 species indicated the occurrence of gain or loss of introns. Pcls occurring in more than or equal to 20% of the 34 species were defined as common insertion sites, while the rest were classified as rare insertion sites. This study identified 15 common Pcls and 28 rare Pcls, of which P1305 was the most universally distributed intron detected in 20 of the 34 species. P612 ranked second and was found in 19 of the 34 species. Rare Pcls (P262, P547, P1193, P1458, P864, P193, P196, P278, P941, P385, P387, P1122, P371, P537, P668, and P821) were present in only 1 of the 34 mitogenomes (Figure 6). Interestingly, several rare Pcls, especially P941, P278, and P369, were identified in species with distant evolutionary relationships, such as *Rhodotorula mucilaginosa*, *G.meredithae*, *Fomitiporia mediterranea*, and *G. calidophilum* from Basidiomycota, suggesting possible gene transfer events.

### 2.7. Comparative Analysis of Cyathus Mitogenomes

A synteny analysis based on five *Cyathus* mitogenomes was conducted to investigate the sequence identity of the complete mitogenome. The results revealed that there were substantial regions of identical sequences in the mitogenomes of *C. striatus* 87405 and AH44044, as well as many regions of high sequence identity between *C. pallidus* QL1 and *C. stercoreus* NPCB004 (Figure 7A). Conversely, fewer identical sequences were identified in *C. jiayuguanensis* compared to the other four *Cyathus* mitogenomes (Figure 7A). The analysis indicated that *C. pallidus* QL1 and *C. stercoreus* NPCB004 share close intragroup evolutionary relationships, whereas *C. jiayuguanensis* is more distantly related. Furthermore, the protein sequence identity of the fourteen concatenated conserved genes between *C. striatus* 87405 and AH44044 exceeded 99%, while it was 97.47% between *C. pallidus* QL1 and *C. stercoreus* NPCB004. In contrast, *C. jiayuguanensis* displayed significantly lower sequence identity with other species, ranging from 81.24% to 84.92% (Figure 7B, Appendix A).

*Cyathus* species had different numbers of un_ORFs encoding reverse transcriptase, LAGLIDADG or GIY-YIG homing endonucleases, and unknown proteins. The *C. striatus* mitogenomes all contained 15 un_ORFs, in which orf230, orf249, orf252, orf261, orf273, orf317, orf 319, orf341, orf344, orf346, orf360, and orf291 encoded LAGLIDADG homing endonucleases; orf183 and orf125 encoded proteins of unknown function; and orf418 encoded a GIY-YIG homing endonuclease.

*Cyathus jiayuguanensis* 765 harbored some un_ORFs homologous to those in *C. striatus*, including orf334, orf353, and orf358, which encode LAGLIDADG homing endonucleases. Two pairs of homologous genes (orf261 and orf261, orf389 and orf366) were distributed in *C. pallidus* QL1 and *C. stercoreus* NPCB004, all encoding LAGLIDADG homing endonucleases. Interestingly, homologous LAGLIDADG homing endonucleases, orf 273 and orf261, all existed in the *cob* gene of *C. striatus* 87405 and AH44044, *C. pallidus* QL1, and *C. stercoreus* NPCB004 (Figure 7C, Appendix A). Therefore, it is speculated that these two un_ORFs are commonly present in the genus *Cyathus*. In addition, we observed varying counts of un_ORFs in different species: 4 in *C. striatus* 87405, 2 in *C. striatus* AH44044, 17 in *C. jiayuguanensis* 765, 9 in *C. pallidus* QL1, and 16 in *C. stercoreus* NPCB004. These results suggest that *C. jiayuguanensis* 765 may have undergone accelerated evolutionary changes compared to other analyzed *Cyathus* species, offering valuable insights into the evolutionary relationships and divergence among *Cyathus* species at the mitogenome level.

### 2.8. Comparative Analysis of Agaricales Mitogenomes

A comparative analysis of the mitochondrial genomes of 34 *Agaricales* species, including five *Cyathus* species, was conducted to understand their mitochondrial features better. The mitogenome sizes showed considerable variation, ranging from 43,328 bp (*Asterophora parasitica*) to 256,807 bp (*Clavaria fumosa*) (Figure 8, Appendix A). The analysis encompassed five genomic components, namely un_ORFs, intronic regions, intergenic regions, RNA regions, and core PCGs, to determine their contribution to mitogenomic differences. The relative content of each component displayed substantial disparity among the mushroom species studied. Core PCGs and RNA regions occupied nearly 50% of mitogenomes in mushrooms such as *Asterophora parasitica*, *Schizophyllum commune*, and *M. tenuissimus*, while accounting for only one-eighth or less in others, including *C. striatus* 87405, *Omphalotus japonicus*, and *Clavaria fumosa*. Intergenic regions also contributed to mitogenome size variation, constituting 46% of *C. jiayuguanensis* 765 and 28% of *Leucoagricus naucinus* mitogenomes. Meanwhile, un_ORFs percentages ranged from 5% to 30% across species (Figure 8, Appendix A).

To explore potential relationships between mitogenomic sizes and genomic contents, we performed correlation analyses. Significant positive correlations were found between mitogenomic lengths and un_ORF sequences (R = 0.92, *p* = 2.2 × 10^−14^), HEGs (R = 0.92, *p* = 1.71 × 10^−14^), intronic regions (R = 0.91, *p* = 4.74 × 10^−14^), and intergenic regions (R = 0.91, *p* = 3.90 × 10^−14^), highlighting their influence on mitogenome size variation (Figure 9A–D, Appendix A). However, RNA regions and core PCG lengths displayed no notable correlation with mitogenomic size (R = 0.06, *p* = 0.73, and R = 0.05, *p* = 0.77, respectively) (Figure 9E,F, Appendix A), indicating minor effects of these components on mitogenome length. Similar positive correlations between intergenic or intronic regions and mitochondrial gene lengths have been reported in Ascomycetes, including Hypocreales order fungi [43] and *Lachancea* [44], *Penicillium* [45], and *Rhynchosporium* genera [46]. The observed differences in mitogenome composition across 34 *Agaricales* mushrooms could be attributed to several factors, including repetitive sequence transfers and gain/loss events of introns and un_ORFs. These mechanisms may explain the observed variations in mitogenome length among these species.

### 2.9. Gene Arrangement of 17 Genes

Apart from the variations in mitogenomic component contents, the gene order—comprising fifteen core proteins, *rns*, and *rnl*—exhibited substantial distinctions across all mitogenomes analyzed (Figure 10). Widespread gene rearrangements, including positional exchanges and gene migrations, occurred in mitogenomes spanning different genera. Several gene rearrangements were also detected in mitogenomes of identical genera, such as *Cyathus*, *Lyophyllum*, and *Pleurotus* (Figure 10). Gene rearrangements within *C. jiayuguanensis* and the other four *Cyathus* mitogenomes, involving inversion, insertion, and transfer events, indicated intricate gene shuffling during the evolutionary development of *Cyathus* species. Conversely, the gene order was maintained in *C. striatus* 87405 and AH44044, *C. pallidus* QL1, and *C. stercoreus* NPCB004, which had the following gene arrangement: *cox1*, *nad6*, *cox2*, *rnl*, *nad2*, *nad3*, *atp6*, *rps3*, *nad1*, *cob*, *nad4*, *rns*, *atp8*, *cox3*, *atp9*, *nad4L*, and *nad5* (Figure 10).

Previous studies have demonstrated that repetitive sequences, particularly in non-coding regions, facilitate the recombination of mitogenomes, thereby increasing the likelihood of gene shuffling events [4]. Therefore, the most plausible explanation for this phenomenon is that *C. jiayuguanensis* possessed the greatest proportion of repetitive (3.10%) and tandem repeat (19.95%) sequences relative to other *Cyathus* species (Appendix A).

### 2.10. Phylogenetic Analyses

Subsequently, we performed maximum likelihood (ML) and Bayesian inference (BI) analysis on the integrated protein sequence dataset (fourteen conserved PCGs), resulting in two phylogenetic trees with consistent and well-supported topologies (Figure 11). The 111 Agaricomycetes species formed 17 major clades of the orders *Agaricales*, Boletales, Russulales, Polyporales, Hymenochaetales, Phallales, Gomphales, Sebacinales, Cantharellales, Tremellales, Trichosporonales, Pucciniales, Microbotryales, Sporidiobolales, Tilletiales, Microstromatales, and Ustilaginales (Appendix A). The 34 *Agaricales* mushrooms were separated into 15 families: Pleurotaceae, Lyophyllaceae, Pluteaceae, Tricholomataceae, Agaricaceae, Hydnangiaceae, Physalacriaceae, Omphalotaceae, Marasmiaceae, Schizophyllaceae, Agaricomycetidae, Hygrophoraceae, Amanitaceae, and Strophariaceae. Moreover, the phylogenetic tree indicated a close relationship between *Cyathus* and the *Laccaria* genus (Hydnangiaceae).

Fungal mitochondrial genomes typically exhibit relatively high variation rates, characterized by disparities in ribosomal RNA sequences and specific protein-coding genes across different families. In light of this, we performed a phylogenetic analysis using the PCG, PCG12, PCGR, and PCG12R datasets derived from 34 *Agaricales* species, as opposed to the amino acids of conserved genes in the 111 Basidiomycete mitogenomes. After subjecting the datasets to heterogeneity analysis and base substitution saturation testing, we constructed eight phylogenetic trees. The *Cyathus* clade displayed identical phylogenetic topologies (*C. jiayuguanensis* + ((*C. pallidus* + *C. stercoreus*) + (*C. striatus* 87405 + *C. striatus* AH44044))) across all four datasets, strongly supported by high BPP (1) and BS (100) values. However, the phylogenetic relationships among *Nidulariaceae*, Strophariaceae, and Hydnangiaceae remained unresolved, as three ML trees and one BI tree supported (*Nidulariaceae* + other species) + (Strophariaceae + Hydnangiaceae), while other trees concurred with the tree based on 111 species and supported other species + (*Nidulariaceae* + (Strophariaceae + Hydnangiaceae)). Furthermore, discrepancies in topologies involving *V. volvacea* and *L. sordlda* were observed across eight trees, with (((((*L. decastes* + *L. shimeji*) + *A. parasitica*) + *M. boudieri*) + *T. bakamatsutake* + *T. matsutake*)) + *L. sordlda*) + *V. volvacea* being the most dominant topology and possessing the highest node support (Figure 12, Appendix A). Based on these results, we confirm the evolutionary position of five *Cyathus* species and believe that the integrated mitochondrial gene data can serve as a robust and reliable genetic marker to characterize the phylogeny of Basidiomycetes.

## 3. Discussion

Compared to what is observed in other eukaryotes, the size of fungal mitogenomes is usually not conserved, and the length of mitogenomes varies considerably even among species of the same genus [3], which may be caused by intron insertions, repetitive sequence distribution, and non-conserved genes caused by the horizontal transfer [47]. The comparative analysis of the mitogenomes of 34 species of *Agaricales* showed that they exhibited some variation in size, ranging from 43,328 bp to 256,807 bp. The mitogenomes of five *Nidulariaceae* species were in the intermediate size range, smaller than those of *A. bisporus* and *Amanita thiersii* and larger than those of *Armillaria sinapina* and *Tricholoma bakamatsutake*. Correlation analysis revealed that the sizes of the un_ORFs, HEGs, intronic region, and intergenic region (Figure 9A–D) all contributed significantly to the mitogenome length (correlation coefficients > 0.9), which suggested the presence of a large number of intron drift events and the acquisition and deletion of un_ORFs in these *Agaricales* species. Furthermore, comparative analyses revealed that the GC skew of the 34 species varied considerably. With the exception of *Nidulariaceae*, most of the species had positive GC skew, which reflected obvious interspecific differences. In animal, plant, and some fungal genomes, GC skew usually carries information on the location of outgoing transcriptional starts, whereas in the bacterial genome, GC skew is closely related to the positioning of the leading and lagging strands [48]. However, there are few studies on mitochondrial GC skew in fungi. Considering the differences in the affinity of DNA double strands for RNA polymerase and related transcription factors, it is speculated that there is a certain relationship between the difference in the GC skew of fungal mitochondrial DNA and the transcription efficiency of mRNA. Therefore, the intrinsic link between GC-skew differences and replication and transcription for species of the order *Agaricales*, especially species of the family *Nidulariaceae*, needs to be further investigated and validated.

Correlation analysis based on the GC content and GC-rich codons showed that the coding genes showed a preference for GC-rich codons among the 34 *Agaricales* species (Figure 4A). Although the GC content of the 34 species varied widely, from 19.74% to 31.73% (Appendix A), the codon bias of these *Agaricales* species showed some conservation (Figure 3B). The reason for this bias may be that changes in GC content occur more in non-coding regions than in coding regions. Furthermore, among the five *Nidulariaceae* species, not only are codon preferences highly conserved, but also encoded tRNA types are completely consistent. All *Nidulariaceae* species contain tRNAs for 20 natural amino acids, with *trnM*, *trnL*, and *trns* all having two copies, and all of their *trnM* anticodons are CAU (Appendix A). Given the correlation between codon bias and tRNA abundance [49,50], it is speculated that *Nidulariaceae* species share similar gene expression patterns.

Mitochondrial gene rearrangements are commonly studied in animals, and these studies suggest that gene rearrangements may be the result of tandem repeat sequence duplication and random loss [51,52]. However, the mechanism of fungal mitochondrial gene rearrangement is less clear. Some studies have shown that the accumulation of tandem repeats and the degree of dispersion of homing endonucleases and tRNAs will lead to changes in gene arrangement [4]. Gene rearrangement analyses have identified large-scale gene rearrangement events in 17 genes in the order *Agaricales* species, and it is thought that such gene rearrangements are caused by these two types of factors. However, in the genus *Cyathus*, the gene order of *C. stercoreus*, *C. pallidus*, *C. striatus* 87405, and AH44044 is highly conserved. In contrast, the gene order of *C. jiayuguanensis* 765 differs from that of other *Cyathus* species. The reason may be that *C. jiayuguanensis* 765 contains the highest proportion of repetitive sequences (Appendix A). This finding suggests the complexity of gene rearrangement in the fungal mitogenome.

A phylogenetic tree constructed from the amino acid sequences of fourteen PCGs in the mitogenomes of 111 Basidiomycete species revealed well-supported evolutionary divergence (Figure 11). In this tree, *C. striatus* 87405 and AH44044 form one of the smallest sister groups, while *C. stercoreus* and *C. pallidus* form the other. This result is consistent with the findings of the synteny analysis (Figure 7A) for the five *Nidulariaceae* species. In the synteny analysis, *C. striatus* 87405 and AH44044 share a large number of sequence identity regions, and *C. stercoreus* and *C. pallidus* also share a large number of highly similar regions (Figure 7A). The consistent results obtained with various analytical methods indicate the dependability of the analytical results. Due to the degeneration of codons, amino acid-based phylogenetic trees cannot accurately capture differences at the codon level. Therefore, to gain a more comprehensive understanding of the phylogenetic topology of the *Nidulariaceae* fungi, we rationally adopted the *Agaricales* dataset to construct eight additional phylogenetic trees. Among them, the *Agaricales* species share a fixed topology with high support. However, it is worth noting that among the eight trees, half of the trees show that the *Nidulariaceae* species form a monophyletic group with the species of Strophariaceae and Hydnangiaceae, while the other half show that the species of *Nidulariaceae* form a parallel group with the species of the Strophariaceae and Hydnangiaceae. Furthermore, the topology of other members of the order *Agaricales* shows differences in different evolutionary trees, such as the evolutionary relationship between *V. volvacea* and *L. sordlda*. Five trees together support the topology of (((((*L. decastes* + *L. shimeji*) + *A. parasitica*) + *M. boudieri*) + *T. bakamatsutake* + *T. matsutake*)) + *L. sordlda*) + *V. volvacea* (Figure 12), and these results were consistent with those of the phylogenetic tree based on fourteen conserved proteins (Figure 11). However, the evolutionary relationship between *V. volvacea* and *L. sordlda* was not the case in the other three phylogenetic trees. These findings provide new insights into the phylogenetic relationships of *Agaricales* fungi and highlight the importance of considering codon-level changes when constructing phylogenetic trees.

## 4. Materials and Methods

### 4.1. Sequencing, Assembly, and Annotation of Mitogenomes

The five strains of *Cyathus* mushrooms used for sequencing were internal transcribed spacer (ITS)-validated, morphologically validated, and preserved in our laboratory. After cultivation in potato dextrose broth (PDB) (Becton, Dickinson, Sparks, NV, USA) medium for ten days, the total DNA of each species was extracted using a Rapid Fungi Genomic DNA Isolation Kit (Sangon Biotech Inc., Shanghai, China) following the manufacturer’s instructions. Whole genome sequencing was performed using a combination of Illumina NovaSeq and Nanopore sequencing technology, achieving 200-fold average genome coverage with a paired-end library. The raw reads (Nanopore) were then de novo assembled using minmap2 (v2.17-r94) (https://github.com/lh3/minimap2, accessed on 18 March 2023) and miniasm (v0.3-r179) (https://github.com/lh3/miniasm, accessed on 18 March 2023). The initial scaffolds were error-corrected based on the Nanopore raw reads using racon (v1.4.20) (https://github.com/isovic/racon, accessed on 18 March 2023). A reliable scaffold was subsequently generated, which was further corrected based on Illumina data using pilon (v1.23) (https://github.com/broadinstitute/pilon, accessed on 18 March 2023). The final assemblies’ quality was assessed via samtools v1.14 (http://www.htslib.org/, accessed on 18 March 2023). Initial annotation of the assembled mitochondrial genome was executed with MFannot v5.4.0 (https://megasun.bch.umontreal.ca/apps/mfannot/, accessed on 18 March 2023), utilizing genetic code 4 for predicting PCGs, tRNA genes, rRNA genes, and open reading frames. To guarantee the validity and dependability of annotation outcomes, it was essential to perform an additional layer of manual proofreading. Verification of tRNA and rRNA annotations was performed using RNAweasel (v5.2.1) (https://github.com/BFL-lab/RNAweasel, accessed on 18 March 2023) and tRNAScan (v2.0.10) (https://www.psc.edu/resources/software/trnascan-se/, accessed on 18 March 2023), respectively. Prediction of tRNA secondary structures was conducted using tRNAscan-SE (v2.0.91) (http://lowelab.ucsc.edu/tRNAscan-SE, accessed on 18 March 2023), and visualization was achieved via VARNA (v1.10) (http://varna.lri.fr/, accessed on 18 March 2023). Intron types were verified through RNAweasel, ensuring type I intron compliance with standard sequence conventions (upstream exons terminating in T and introns closing with G). To confirm the accuracy of *rns*, *rnl*, PCGs, and intron insertion sites, MAFFT (v7.453) (https://mafft.cbrc.jp/alignment/software/, accessed on 21 March 2023) and BLAST (v2.13) (https://blast.ncbi.nlm.nih.gov/Blast.cgi, accessed on 21 March 2023) analysis were employed. ORF Finder online software (v1.8) (https://www.ncbi.nlm.nih.gov/orffinder/, accessed on 21 March 2023) was utilized to search for unannotated open reading frames within intergenic or intron regions exceeding 300 bp. Conjoint analysis with Blastn and BlastP was conducted to determine ORF start sites within introns and ascertain their respective functions. Lastly, OGDraw (v1.2) (https://chlorobox.mpimp-golm.mpg.de/OGDraw.html, accessed on 21 March 2023) enabled the generation of graphical maps for five complete mitogenomes.

### 4.2. Sequence Analysis of Mitogenomes

In this extensive study, we analyzed the mitogenomes of 34 distinct *Agaricales* species, focusing on features such as nucleotide composition, codon usage bias, and evolutionary pressures. We assessed the GC content, GC skew, and AT skew utilizing a Python script, where GC content is defined as (G + C)/(G + C + A + T), GC skew as (G − C)/(G + C) and AT skew as (A − T)/(A + T). We determined the RSCU values of mitochondrial PCGs, core PCGs, and un_ORFs using an online tool (http://cloud.genepioneer.com:9929/#/login, accessed on 25 March 2023). We performed t-distributed stochastic neighbor embedding (t-SNE) analyses based on RSCU values with the Rtsne R package (v0.15) (https://github.com/jkrijthe/Rtsne, accessed on 25 March 2023).

Fifteen core protein-coding genes (*atp6*, *atp8*, *atp9*, *cob*, *cox1*, *cox2*, *cox3*, *nad1*, *nad2*, *nad3*, *nad4*, *nad4L*, *nad5*, *nad6*, *and rps3*) from 34 mitogenomes were individually aligned using MAFFT (v7.453). Synonymous (Ks) and non-synonymous (Ka) substitution rates were calculated using DnaSP (v6.12.03) (http://www.ub.edu/dnasp/, accessed on 26 March 2023). The Ka/Ks ratio was subsequently computed to explore evolutionary pressure. To analyze the genetic distance of the core PCGs, MEGA 11 (v11.0.13) (https://megasoftware.net/, accessed on 26 March 2023) was used to align the identical type of PCGs and calculate Kimura 2-parameter (K2P) distances between pairs among 15 PCGs.

To investigate the presence of intra-genomic duplications of large fragments or interspersed repeats within the five *Cyathus* mtDNAs, we performed BLASTn analysis (e-value 10^−10^) by comparing the mitogenomes against themselves. We utilized Tandem Repeat Finder (v4.07b) (https://tandem.bu.edu/trf/submit_options, accessed on 26 March 2023) with default parameters to identify tandem repeats longer than 10 bp.

### 4.3. Comparative Analyses and Intron Analysis of Mitogenomes

Synteny across the entire genomes of the five *Cyathus* species was assessed using AliTV (https://alitvteam.github.io/AliTV/d3/AliTV.html, accessed on 30 March 2023), with all five mitogenomes starting from the *cox1* gene. Protein sequence identities for fourteen concatenated conserved proteins among five *Cyathus* species were determined using Clustal Omega (https://www.ebi.ac.uk/Tools/msa/clustalo, accessed on 2 April 2023). Homologous un_ORFs (nucleotide sequence identity > 80%) shared by five *Cyathus* species were identified using the same tool.

For 34 *Agaricales* species examined, the lengths of core PCGs, RNA regions, intergenic regions, intronic regions, HEGs, and un_ORFs were compared. A stacked bar graph illustrating the length and compositional distribution of the mitogenomes was generated using the ggplot2 package (v.3.4.2) (https://github.com/tidyverse/ggplot2, accessed on 25 March 2023). If a gene was situated within an intron, the intronic sequence length was calculated by subtraction of the size of the gene. Mitogenome size and the associations between its six compositions (core PCGs, RNA regions, intergenic regions, intronic regions, homing endonuclease genes HEGs, and un_ORFs) were evaluated using the cor.test() function in R. Previous reports were consulted for intron analysis of the *cox1* genes of 34 species [41,42].

### 4.4. Phylogenetic Inference

To explore the evolutionary relationships within the Basidiomycota phylum, we analyzed the mitogenomes of various species and constructed a phylogenetic tree based on the amino acid sequences of fourteen conserved protein-coding genes in 111 Basidiomycota species. Amino acid alignments were performed using MAFFT (v 7.453) with default parameters. Subsequently, the separate protein sequences were merged into combined datasets using PhyloSuite (v1.2.2) (https://github.com/dongzhang0725/PhyloSuite, accessed on 5 April 2023). Bayesian inference (BI) analysis was conducted using MrBayes (v3.2.7) (http://nbisweden.github.io/MrBayes/, accessed on 5 April 2023), employing two runs of four chains and 2 × 10^6^ generations, along with a 100-sample frequency and 25% burn-in fraction. Iterations were considered to have reached convergence when the estimated sample size (ESS) exceeded 100 and the potential scale reduction factor (PSRF) approached 1. Maximum likelihood (ML) analysis was performed with IQ-TREE (v2.0.3) (http://www.iqtree.org/, accessed on 5 April 2023) under the LG+F+I+G4 model, incorporating 5000 bootstrap replicates and the Shimodaira–Hasegawa-like approximate likelihood ratio test.

We further established four datasets derived from fourteen conserved PCGs and ribosomal large and small subunit genes (*rnl* and *rns*) of 34 *Agaricales* species: (1) PCG—concatenated sequences of fourteen conserved PCGs; (2) PCG12—combined 1st and 2nd codon positions of fourteen conserved genes; (3) PCGR—concatenated fourteen conserved PCGs and rRNA; and (4) PCG12R—combined 1st and 2nd codon positions of fourteen conserved genes and rRNA. Initially, individual sequences were aligned with MAFFT (v7.453) using default settings. Gblocks and Trimal were applied to remove poorly aligned regions, followed by the use of PhyloSuite to concatenate protein sequences and coding genes. Partition Finder (v2.1.1) (https://github.com/brettc/partitionfinder, accessed on 5 April 2023) was deployed to identify the optimal partitioning scheme and evolutionary models for the four combined gene datasets, guided by the greedy algorithm and the AICc criterion. Base substitution saturation and sequence heterogeneity were assessed using AliGROOVE (v1.07) (https://github.com/PatrickKueck/AliGROOVE, accessed on 5 April 2023) and DAMBE7 (http://dambe.bio.uottawa.ca/DAMBE/dambe_install_win.aspx, accessed on 5 April 2023) for the four datasets. Phylogenetic trees were constructed via Bayesian inference (BI) and maximum likelihood (ML) methods, utilizing MrBayes (v3.2.7) and IQ-TREE (v2.0.3), respectively. Finally, the resulting phylogenetic trees were visualized using Figtree (v1.4.4) (https://github.com/rambaut/figtree/releases, accessed on 5 April 2023).

### 4.5. Data Availability

The complete mitogenomes of *C. striatus* 87405, *C. striatus* AH44044, *C. jiayuguanensis* 765, *C. pallidus* QL1, and *C. stercoreus* NPCB004 were deposited in the GenBank database under the accession numbers OP693452, OP693453, OP693454, OQ466117, and OQ466118, respectively. Five *Agaricales* mitogenomes (*A. bisporus*, *Flammulina velutipes*, *Omphalotus japonicus*, *S*. *commune*, and *S. rugosoannulata*) from the NCBI database were re-annotated using the same pipeline utilized for five *Cyathus* mitogenomes. These re-annotation files are available in the Appendix A.

## 5. Conclusions

In this study, we de novo assembled and annotated five mitochondrial genomes from the genus *Cyathus* (*C. striatus* 87405, *C. striatus* AH44044, *C. jiayuguanensis* 765, *C. pallidus* QL1, and *C. stercoreus* NPCB004), generating high-quality data for future research. Comparative analysis of the mitogenomes of 34 *Agaricales* species revealed conservation in codon usage preferences and Ka/Ks ratios for most conserved genes. However, significant differences in mitochondrial genome length and intron insertion sites were observed. These genome length divergences were attributed to un_ORFs, HEGs, and intronic and intergenic regions. The *Agaricales* species exhibited a plethora of homing endonucleases and diverse intron sites, indicating that large-scale intron gain or loss events have occurred during evolution. In addition, gene rearrangements were evident among the 34 species, indicating substantial gene order variation, even within the same genera. Phylogenetic trees constructed using amino acids from fourteen conserved genes and four datasets (PCG, PCG12, PCGR, and PCG12R) revealed the evolutionary position of *Cyathus* fungi within *Agaricales* and confirmed their monophyly. In particular, subspecies 87405 and AH44044 of *C. striatus* showed high intersubspecific homology and sequence identity, whereas *C. pallidus* QL1 and *C. stercoreus* NPCB004 showed significant interspecific similarity. These results are consistent with the affinities shown in the five phylogenetic trees.

In conclusion, the mitogenome analyses not only reflect the presence of gene rearrangements and abundant intron dynamics in the mitogenome of the genus *Cyathus*, but also clarify the phylogenetic position of *Cyathus* within the family *Nidulariaceae*. These results provide valuable information for species identification in the genus *Cyathus* and offer new perspectives for understanding the evolutionary status of *Cyathus* species.

## Figures and Tables

**Figure 1 ijms-24-12599-f001:**
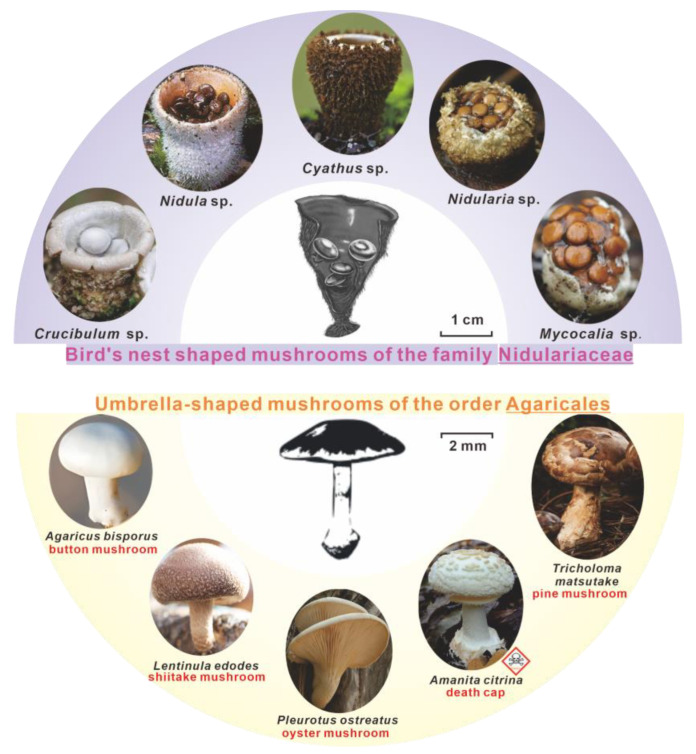
Comparison of the fruiting body morphology of *Nidulariaceae* and other *Agaricales* outside *Nidulariaceae*.

**Figure 2 ijms-24-12599-f002:**
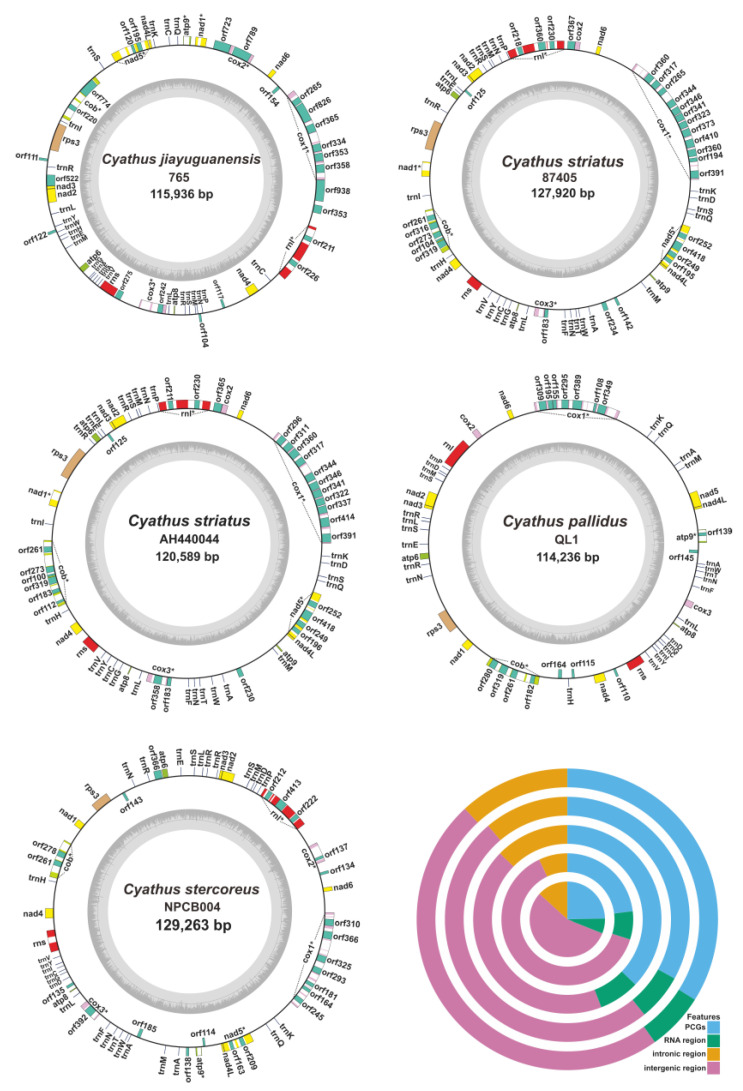
Circular maps of the mitogenomes of five *Cyathus* species. Genes are shown by blocks of different colors. The doughnut chart represents different compositions of the total mitochondrial genomes of the five *Cyathus* species. From the inner doughnut to the outer ring, *C. stercoreus* NPCB004, *C. pallidus* QL1, *C. jiayuguanensis* 765, *C. striatus* 87405, and *C. striatus* AH44044 are sequenced. The symbol * indicates that the corresponding gene contains an intron (introns).

**Figure 3 ijms-24-12599-f003:**
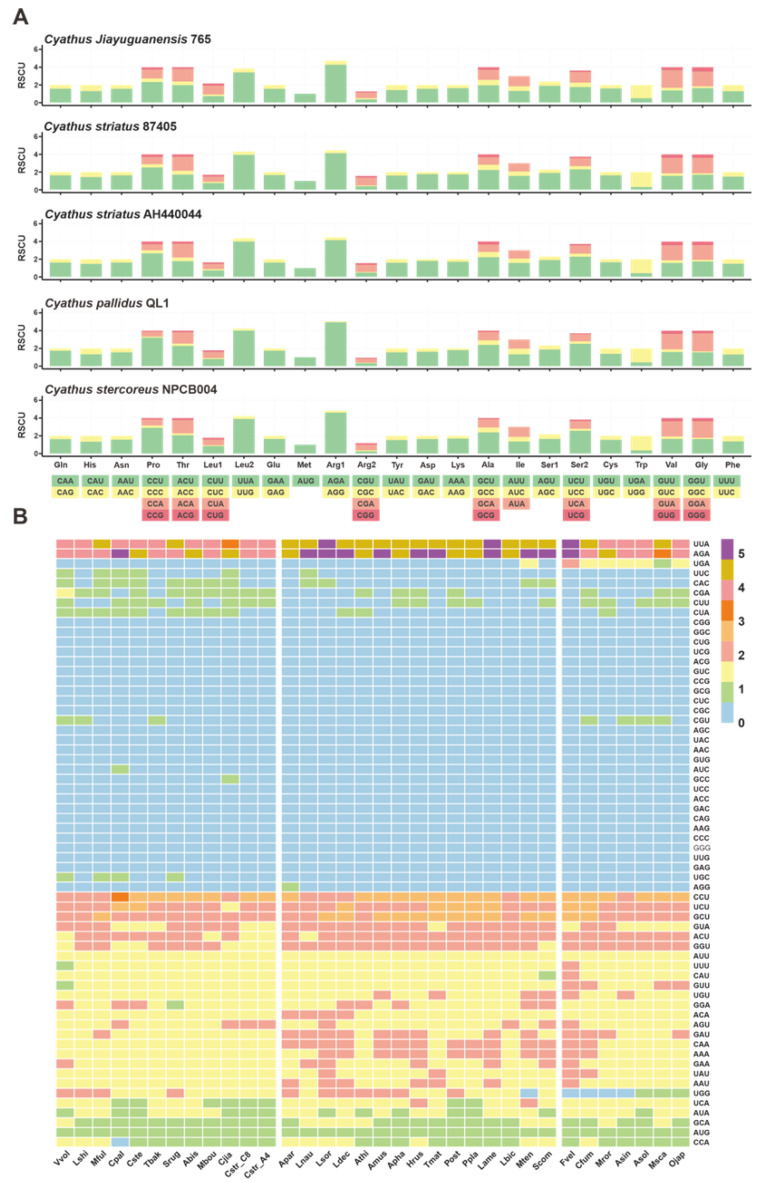
(**A**) Stacked column plots of the RSCU of the mitochondrial genomes of five *Cyathus* species. (**B**) Heatmap of the RSCU of the mitogenomes of 34 *Agaricales* species.

**Figure 4 ijms-24-12599-f004:**
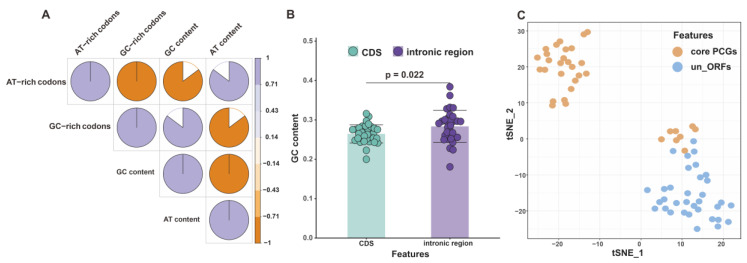
(**A**) Pearson correlation matrix between the usage of AT-rich codons, GC-rich codons, AT content, and GC content in 34 *Agaricales* species (**B**) GC content of mitochondrial CDS and intronic regions from 34 *Agaricales* species. (**C**) T-SNE analysis of the RSCU in mitochondrial core PCGs and un_ORFs.

**Figure 5 ijms-24-12599-f005:**
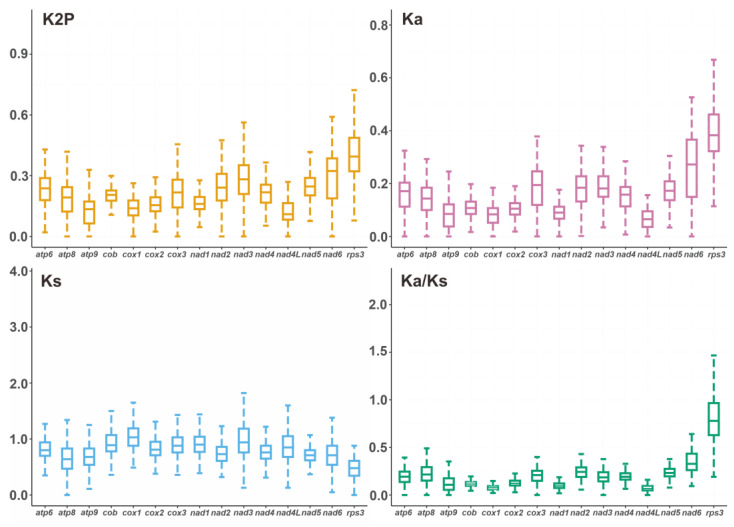
Genetic analysis of 15 protein-coding genes conserved in the 34 *Agaricales* mitogenomes. K2P, Kimura 2-parameter distance; Ka, the mean number of non-synonymous substitutions per non-synonymous site; Ks, the mean number of synonymous substitutions per synonymous site.

**Figure 6 ijms-24-12599-f006:**
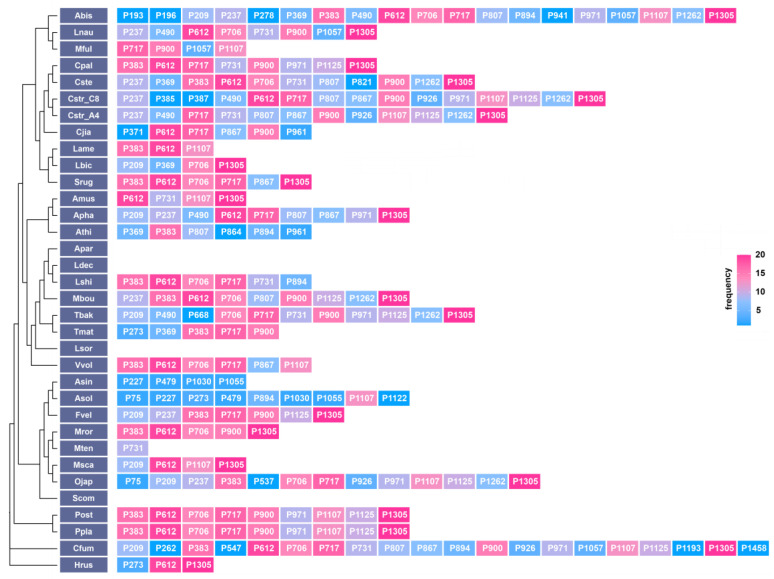
Intron insertion sites in the cox1 gene of 34 *Agaricales* species. The phylogenetic tree was constructed using amino acids from fourteen conserved PCGs.

**Figure 7 ijms-24-12599-f007:**
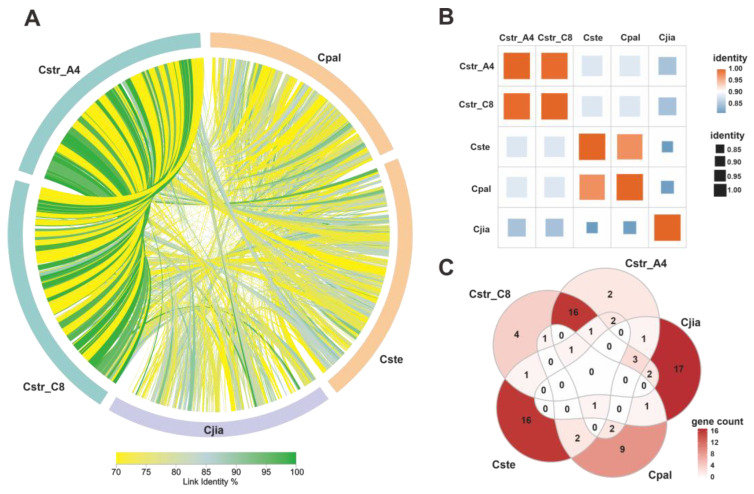
(**A**) Mitogenome synteny among the five *Cyathus* species, links with identity > 70% are shown. (**B**) The protein sequence identities of fourteen concatenated conserved genes among five *Cyathus* species. (**C**) Homologous un_ORFs (nucleotide sequence identity > 80%) shared among five *Cyathus* species.

**Figure 8 ijms-24-12599-f008:**
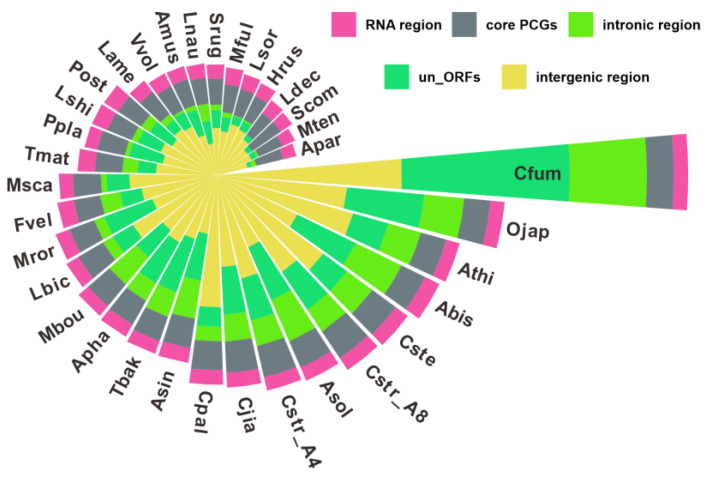
The lengths of RNA region, core PCGs, intronic region, un_ORFs, and intergenic region in 34 Nidulariaceae species.

**Figure 9 ijms-24-12599-f009:**
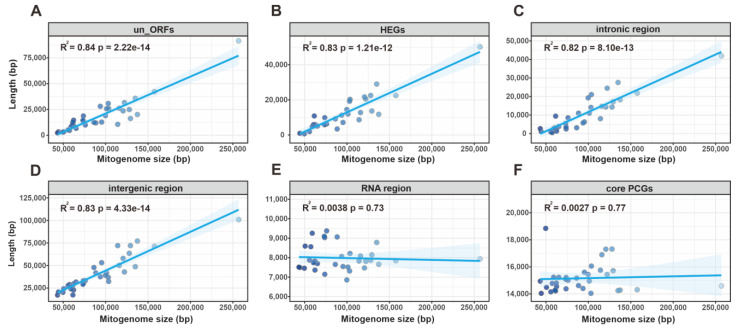
Correlation analysis between the lengths of mitogenomes and the lengths of un_ORFs (**A**), HEGs (**B**), intronic region (**C**), intergenic region (**D**), RNA region (**E**), and core PCGs (**F**).

**Figure 10 ijms-24-12599-f010:**
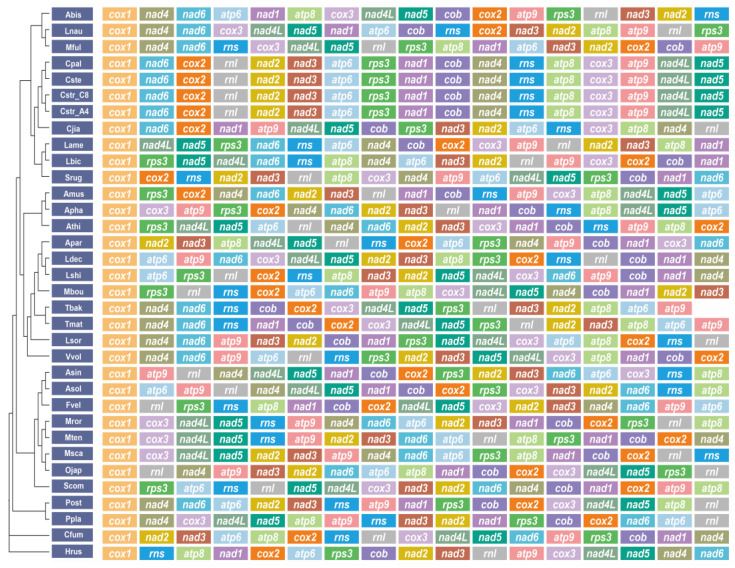
Gene order analyses for 34 *Agaricales* mitogenomes. The same gene is represented by the same background color. All genes (fifteen core PCGs and two ribosomal RNAs) are shown in the order of their appearance in the mitogenome, starting with *cox1*.

**Figure 11 ijms-24-12599-f011:**
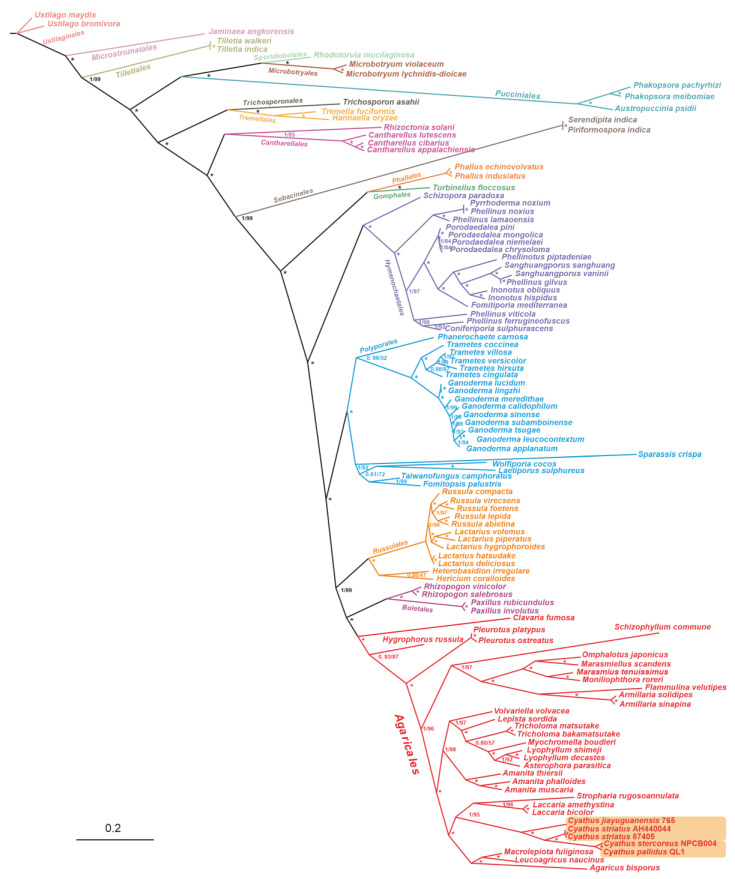
Phylogeny of 111 Basidiomycetes species based on amino acids of fourteen conserved PCGs. Values before the slash at the clades indicate Bayesian posterior probabilities (BPPs), while those after the slash at the clades indicate bootstrap (BS). The asterisk at the clade indicates a BPP value of 1 and a BS value of 100. The area where the *Cyathus* species are located is marked with an orange background.

**Figure 12 ijms-24-12599-f012:**
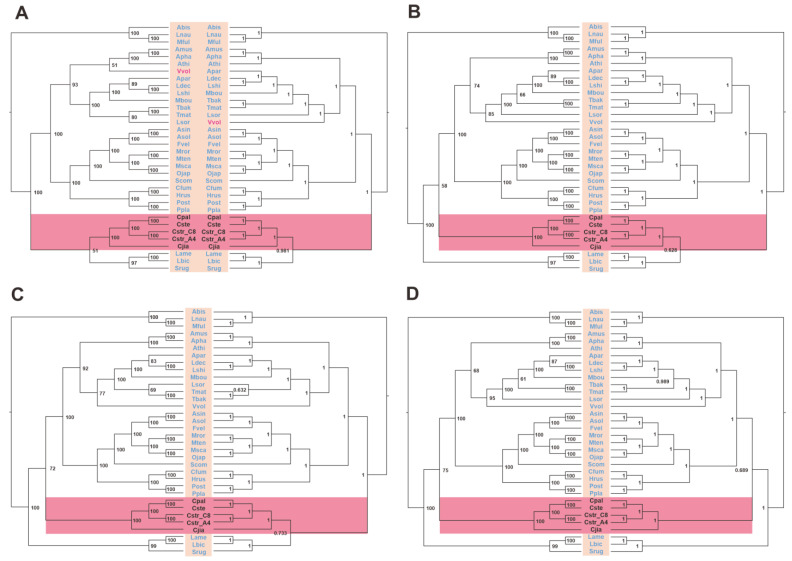
BI and ML trees conducted based on the four datasets: D1: PCG (**A**); D2: PCG12 (**B**); D3: PCGR (**C**); D4: PCG12R (**D**). The relatively conserved tribes are in different color blocks.

## Data Availability

All data generated or analyzed during this study are included in this published article and its additional files.

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
