# Peer review of "The First Five Mitochondrial Genomes for the Family Nidulariaceae Reveal Novel Gene Rearrangements, Intron Dynamics, and Phylogeny of Agaricales"

_ijms, 2023, doi:10.3390/ijms241612599_

Round 1

Reviewer 1 Report

The paper is rather long. But it seems to be O.K. However, I am no expert of all sections of this manuscript. I made several small remarks/corrections as commentaries.

The English seems to be O.K.

Author Response

Q1. The paper is rather long. But it seems to be O.K. However, I am no expert of all sections of this manuscript. I made several small remarks/corrections as commentaries.

A1:Thank you for your review and helpful suggestions. The length of the manuscript seems to exceed the usual length of a manuscript due to the first systematic study of the mitochondrial genome of the family Nidulariaceae, where we found several scientifically significant results. Although these results have been rounded off according to their importance, the results presented are still very long. Thank you for your understanding.

Q2.Comments on the Quality of English Language:The English seems to be O.K.

A2: Thank you for recognizing English spelling. We take the opportunity to improve English writing to make it more readable and scientific.

Q3. L18:are there cortresponding investigations on other Nidulariaceae?

A3:This study is the first investigation of the mitochondrial genome of Nidulariaceae, to our knowledge.

Q4. L44: remove ,

A4: Done.

Q5. L55:and paricularly of DNA parts

A5: The appropriate supporting literatures has been added, and your suggestions are appreciated.

Q6. L63 L166 L331: species, not mushrooms

A6: Modified.

Q7. L66 L140: why bold-faced? why black type?

A7: It's just a typographical convention, and bolding as a marker helps with quick orientation. Of course, the decision to bold or not will ultimately depend on the typographical arrangements at the time of possible subsequent publication.

Q8. L71:add Reference

A8: The URL for supporting information has been added, thank you for your suggestion.

Q9. Figure 1:current name is Lentinula edodes

A9: Thank you for your careful observation, I am sorry for this oversight and have corrected it as you suggested.

Q10. L113, L216: add Reference

A10: The appropriate supporting literatures have been added, and your suggestions are appreciated.

Q11. L142 : colour, British or American English?

A11: Thank you for your interest in Word Spelling Style. The vocabulary throughout the manuscript favors British English. We have changed color to colour to ensure a consistent style of vocabulary.

Q12. L171 L176: add species

A12: It has been added, thanks for the correction.

Q13. L215: why are the following lines intended?

A13: We are very sorry to say that this was due to a typographical oversight when we were writing the manuscript. Thank you for pointing this out and we have corrected it.

Q14. L250 L251: unclear

A14: Thank you for your careful review, which was caused by improper typesetting of the manuscript. As you know, IJMS requires that manuscripts be submitted using a specific template. This error occurred when we transferred the content from the original manuscript to this one.

Q15. L287: small letter

A15: Thank you for the careful reminder. As the species name should be in lower case here, we have corrected it.

Q16. L312 : italics

A16: Thanks for the careful reminder that species and genus names should be in italics, which we have changed.

Q17. L339: M = Marasmius?

A17: Yes. This is not the first appearance of the species (Latin name), so the species name is used as an abbreviation. The full name of the species has appeared before.

Q18 L356: 1. Lachancea 2. italics!

A18: The words “Lachancea”, “Penicillium”, and “Rhynchosporium” have been italicized, which we have changed.

Q19. L506: the section Methods should be placed behind the Introduction

A19: Thank you for your constructive suggestions. However, IJMS has its own specific template requirements. The order of the sections is now according to the template requirements.

Q20. L508, L509: 1.use full words when mentioned for the first time! internal transcribed spacer (ITS)2. potato dextrose agar /PDA) mention concentration

A20: Relevant abbreviations have been changed to all caps and relevant details have been added.

Q21 L522: ‘PCG’ explain

A21: PCG indicates protein-coding gene, and this full name appears where it is first mentioned, on line 36.

Q22 L522: ‘t’ explain

A22: The 'tRNA' is a proper noun denoting transporter RNA, where 't' means transporter.

Q23 L522: ‘r’ explain

A23: R refers to ribosomal, and rRNA is a specialized term.

Q24 L532: ‘MAFFT’ explain

A24: 'MAFFT' is the name of a software program that is used to align sequences

Q25 L533: ‘BLAST’ explain.

A25: 'BLAST', known as 'Basic Local Alignment Search Tool', is a search engine under the 'NCBI' website. This type of abbreviation is common and allowed in biology related academic papers.

Q26. L534: ‘ORF’ explain.

A26: ORF indicates open reading frames, and this full name appears where it is first mentioned, on line 116.

Q27 L538: ‘OGDraw’ explain.

A27: 'OGDraw' is a drawing website that we can use to map the mitochondrial genome.

Q33 L542: ‘In this extensive study, we’ change to be W.

A33: Modified.

Q28. L549: ‘RSCU’ explain.

A28: RSCU means relative synonymous codon usage, and this full name appears where it is first mentioned, on line 193.

Q29. L568: ‘five’ remove.

A29: It has been deleted.

Q30. L606: ‘AliGROOVE’ explain

A30: 'AliGROOVE' is a software can be used to analyse sequence heterogeneity of concatenated supermatrices

Q31. L614, L617, L618: Change ‘C’ to ‘Cyathus’, Change ‘A’ to ‘Agaricus’, Change ’S’ to ‘Schizophyllum’, Change ’S’ to ‘Stropharia’

A31: As is customary in scientific and technical writing, it is indeed necessary to give the full name of a species the first time it is mentioned. However, since these species were mentioned earlier in the paper, we will use abbreviations rather than full names to indicate.

Q32 L614: 'C. striatus 87405 'wild species?

A32: Yes. It was isolated from the wild and identified by ITS and morphology as C. striatus, no. 87405.

Q33. L617 L618: ‘A. bisporus’ add ‘J.E.Lange’, ‘Flammulina velutipes’ add author, ‘Omphalotus japonicus’ add author, ‘S.commune’ add author, ‘S. rugosoannulata’ add author

A33: Thank you for your kind reminder. The species names in the manuscript are written in a way that is consistent with the norms of academic papers related to biology. In addition, even for specific species, the GenbanK accession numbers given in our Supplementary Material provide clues to the details of the species names.

Reviewer 2 Report

English is fine, easy to understand

I think the bioinformatic analyses are as correct as they can be.
I just wonder why the Kimura-2 model was used in MEGA 11 "to align the identical type of PCGs and calculate Kimura-2 parameter (K2P) distances between pairs among 15 PCGs." (lines 556-557), rather than the "Tamura-3 parameter model" with three parameters

Do these conclusions from research enable easier identification of commonly protected fungal species? A quick and effective tool is needed to determine which fungus is in a rotting tree trunk or in overgrown plant debris in the soil. Do the new discoveries help to understand the phylogeny (evolution), but also the origin of the fungi found in a particular area? I mean an introduction to a specific region/forest.

Author Response

Q1: Comments and Suggestions for Authors:English is fine, easy to understand.

Comments on the Quality of English Language:I think the bioinformatic analyses are as correct as they can be.

A1: Thank you for recognizing it.

Q2: I just wonder why the Kimura-2 model was used in MEGA 11 "to align the identical type of PCGs and calculate Kimura-2 parameter (K2P) distances between pairs among 15 PCGs." (lines 556-557), rather than the "Tamura-3 parameter model" with three parameters.

A2: The conventional analytical approach, in almost all reported studies, involves aligning and calculating K2P distances using the Kimura-2 model. However, the Tamura-3 parameter model has rarely been employed in these researches.

Q3: Do these conclusions from research enable easier identification of commonly protected fungal species? A quick and effective tool is needed to determine which fungus is in a rotting tree trunk or in overgrown plant debris in the soil.

A3: Nidulariaceae species are morphologically easily distinguished from other members of the order Agaricales, and one of the aims of the present study was to differentiate these species at the mitochondrial level.

Q4: Do the new discoveries help to understand the phylogeny (evolution), but also the origin of the fungi found in a particular area? I mean an introduction to a specific region/forest.

A4: These new findings contribute to the understanding of the phylogeny and evolution of fungi in the order Agaricales. Whether there is a contribution to the understanding of the origin of fungi in a specific region cannot be assessed, as the data for the species of interest used for analysis in this work are not geographically specific. However, the analytical methods and ideas used are applicable to fungal phylogeny and evolution in all environments.